# Deep Subspace Clustering with Data Augmentation

**Mahdi Abavisani**
Rutgers University
New Brunswick, NJ
mahdi.abavisani@rutgers.edu

**Alireza Naghizadeh**
Rutgers University
New Brunswick, NJ
ar.naghizadeh@rutgers.edu.

**Dimitris N. Metaxas**
Rutgers University
New Brunswick, NJ
dnm@cs.rutgers.edu

**Vishal M. Patel**
Johns Hopkins University
Baltimore, MD
vpatel36@jhu.edu

## Abstract

The idea behind data augmentation techniques is based on the fact that slight changes in the percept do not change the brain cognition. In classification, neural networks use this fact by applying transformations to the inputs to learn to predict the same label. However, in deep subspace clustering (DSC), the ground-truth labels are not available, and as a result, one cannot easily use data augmentation techniques. We propose a technique to exploit the benefits of data augmentation in DSC algorithms. We learn representations that have consistent subspaces for slightly transformed inputs. In particular, we introduce a temporal ensembling component to the objective function of DSC algorithms to enable the DSC networks to maintain consistent subspaces for random transformations in the input data. In addition, we provide a simple yet effective unsupervised procedure to find efficient data augmentation policies. An augmentation policy is defined as an image processing transformation with a certain magnitude and probability of being applied to each image in each epoch. We search through the policies in a search space of the most common augmentation policies to find the best policy such that the DSC network yields the highest mean Silhouette coefficient in its clustering results on a target dataset. Our method achieves state-of-the-art performance on four standard subspace clustering datasets. The source code is available at: https://github.com/mahdiabavisani/DSCwithDA.git.

## 1 Introduction

Recent advances in technology have provided massive amounts of complex high dimensional data for computer vision and machine learning applications. High dimensionality has adverse effects, including confusion of algorithms with irrelevant dimensions and *curse of dimensionality* as well as increased computation time and memory. This motivates us to explore techniques for representing high-dimensional data in lower dimensions. In many practical applications such as face images under various illumination conditions [1] and hand-written digits [2], high-dimensional data can be represented by union of low-dimensional subspaces. The subspace clustering problem aims at finding these subspaces. In particular, the objective of subspace clustering is to find the number of subspaces, their basis and dimensions, and assign data to these subspaces [3].

Conventional subspace clustering algorithms assume that data lie in linear subspaces [4, 5, 6, 7, 8]. In practice, however, many datasets are better modeled by non-linear manifolds. To deal with this issue, many works have incorporated projections and kernel tricks to express non-linearity [9, 10, 7, 11, 12, 13]. Recently, deep subspace clustering (DSC) methods [14, 15, 16, 17, 18] have been

proposed which essentially learn unsupervised nonlinear mappings by projecting data into a latent space in which data lie in linear subspaces. Deep subspace clustering networks have shown promising performances on various datasets.

Deep learning techniques are prone to overfitting. Data augmentation is often presented as a type of regularization to mitigate this issue [19, 20]. While data augmentation for deep learning-based methods have proven to be beneficial, the current framework of DSC networks is unable to take the full advantage of data augmentation. In this work, we modify the DSC framework and propose a model that can incorporate data augmentation into DSC.

An important difference between data augmentation in subspace clustering and data augmentation in supervised tasks is the fact that as opposed to supervised tasks, we do not have ground-truth labels for the existing samples in the subspace clustering algorithms. Corresponding to the fact that objects remain the same even if we slightly transform them, in supervised deep learning models, transformations of an existing sample are trained to be predicted with a consistent label similar to the ground-truth label of the original sample. How can one convey such property in an unsupervised subspace clustering task, where the data does not have the ground-truth labels?

A DSC model should favor functions that give consistent outputs for similar data points with a slight difference in their percept. To achieve this, we optimize a consistency loss that is based on temporal ensembling. We input plausible transformations of existing samples into the model and require the autoencoders of the model to map the transformations to consistent subspaces similar to the subspace of the original data.

Efficient augmentation policies improve the performance of the deep networks. However, not all the image transformations construct efficient augmentation policies. Efficient augmentation policies can be different from a dataset to another [21, 22, 23]. In supervised applications, the validation set is often used to manually search among transformations such as rotation, horizontal flip, or translation by a few pixels to find efficient augmentations. Manual augmentation needs prior knowledge and expertise, and it can only search among a handful of pre-defined trials. Some methods automate this search for classification networks [21, 24, 25]. However, these methods are only designed for the classification task and cannot be applied to the task of subspace clustering. This is because we do not have a validation or training set in subspace clustering. We overcome this issue by providing a simple yet effective method for finding efficient augmentation policies using a greedy search and use mean Silhouette scores to evaluate the effect of different augmentation policies on the performance of our proposed model.

## 2  Related Work

**Clustering Methods with Augmentation.**  A recent method proposes a technique for deep embedded clustering algorithms with augmentations [26, 27]. In the pre-training stage they use augmentations in training autoencoders, and in the fine-tuning stage they encourage the augmented data to have the same centroid as their corresponding data. To the best of our knowledge, we are the first to propose an augmentation framework for deep subspace clustering algorithms.

**Self-supervision with Consistency Loss.**  The idea of learning consistent features for different transformations of unlabeled data has been used in a number of works largely in the semi-supervised and self-supervised learning literature [28, 29, 30, 31, 32, 33].

**Self-expressiveness Models in Subspace Clustering.**  Let $\mathbf{X} = [\mathbf{x}_1, \cdots, \mathbf{x}_N] \in \mathbb{R}^{D \times N}$ be a collection of $N$ signals $\{\mathbf{x}_i \in \mathbb{R}^D\}_{i=1}^N$ drawn from a union of $n$ linear subspaces $\mathcal{S}_1 \cup \mathcal{S}_2 \cup \cdots \cup \mathcal{S}_n$. Given $\mathbf{X}$, the task of subspace clustering is to find sub-matrices $\mathbf{X}_\ell \in \mathbb{R}^{D \times N_\ell}$ that lie in $\mathcal{S}_\ell$ with $N_1 + N_2 + \cdots + N_n = N$.

Due to their simplicity, theoretical correctness, and empirical success, subspace clustering methods that are based on *self-expressiveness property* are very popular [34]. Self-expressiveness property can be stated as

$$\mathbf{X} = \mathbf{XC} \ \ s.t \ \ \mathrm{diag}(\mathbf{C}) = \mathbf{0}, \tag{1}$$

where $\mathbf{C} \in \mathbb{R}^{N \times N}$ is the coefficient matrix. There may exist many coefficient matrices that satisfy the condition in (1). Among those, *subspace preserving* solutions are especially of interest to self-expressiveness based subspace clustering methods. Subspace preserving property states that if an

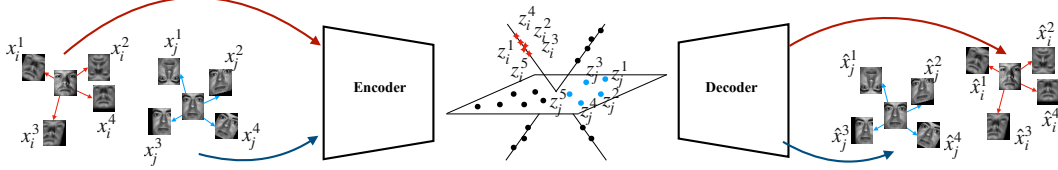

Figure 1: An overview of the proposed deep subspace clustering networks with data augmentation. The existing data points $x_i$ and $x_j$ are transformed into $x_i^t$ and $x_j^t$ in each iteration by an augmentation policy. However, the autoencoder learns to keep their latent space features within consistent subspaces.

element in $\mathbf{C}$ is non-zero, the two data points in $\mathbf{X}$ that correspond to this coefficient are in the same subspace.

Self-expressiveness based methods combine these two properties and solve a problem of the form:

$$\min_{\mathbf{C}} \mathcal{L}_{\text{S.E.}}(\mathbf{C}, \mathbf{X}) + \lambda_1 \mathcal{L}_{\text{S.P.}}(\mathbf{C}), \tag{2}$$

where $\lambda_1$ is a regularization constant, $\mathcal{L}_{\text{S.E.}}$ and $\mathcal{L}_{\text{S.P.}}$ impose the self-expressiveness and subspace-preserving properties, respectively. Most of the linear methods use $\mathcal{L}_{\text{S.E.}}(\mathbf{C}, \mathbf{X}) = \|\mathbf{X} - \mathbf{X}\mathbf{C}\|_F^2$. However, for $\mathcal{L}_{\text{S.P.}}(\mathbf{C})$, different methods use various regularizations, including $\ell_1$-norm, $\ell_2$-norm and nuclear norm [4, 34, 35].

In recent years, deep neural network-based extensions were introduced to self-expressiveness based models [14, 15, 16, 17]. For these methods, $x_i$s do not need to be drawn from a union of linear subspaces. Instead, they use autoencoder networks to map the data points to a latent space where data points lie into a union of linear subspaces and exploit the self-expressiveness and subspace-preserving properties in the latent space. Let $\mathbf{Z} \in \mathbb{R}^{d \times N}$ be the latent space features developed by the encoder in the autoencoders. Deep subspace clustering networks solve a problem of the form:

$$\min_{\Theta} \mathcal{L}_{\text{S.E.}}(\mathbf{C}, \mathbf{Z}) + \lambda_1 \mathcal{L}_{\text{S.P.}}(\mathbf{C}) + \lambda_2 \mathcal{L}_{\text{Rec.}}(\mathbf{X}, \hat{\mathbf{X}}), \tag{3}$$

where $\lambda_1$ and $\lambda_2$ are regularization constants, $\Theta$ is the union of trainable parameters, $\hat{\mathbf{X}}$ is the reconstruction of $\mathbf{X}$ and the output of the decoder, and $\mathcal{L}_{\text{Rec.}}(\mathbf{X}, \hat{\mathbf{X}}) = \|\mathbf{X} - \hat{\mathbf{X}}\|_F^2$ is the reconstruction loss in training the autoencoder. Once a proper $\mathbf{C}$ is found from (2) or (3), spectral clustering methods [36] are applied to the affinity matrix $\mathbf{W} = |\mathbf{C}| + |\mathbf{C}|^T$ to obtain the segmentation of the data $\mathbf{X}$.

## 3 Deep Subspace Clustering Networks with Data Augmentation

The human brain considers an object to remain the same, even if the percept changes slightly. Correspondingly, when data augmentation is used in supervised deep learning models, transformations of existing samples are trained to predict consistent labels similar to the ground-truth label of original samples. Conveying the same insight, we argue that a DSC model should favor functions that give consistent outputs for similar data points. We approach this property by keeping the estimated subspace membership of data points consistent when an augmentation policy is applied to them. During the training process, we smooth the predictions for the subspace memberships via temporal ensembling of estimated affinity matrices from previous iterations.

Let $\mathbf{X}^t = [\mathbf{x}_1^t, \cdots, \mathbf{x}_N^t] \in \mathbb{R}^{D \times N}$ be the transformed version of $N$ existing data points $\mathbf{X} = [\mathbf{x}_1, \cdots, \mathbf{x}_N] \in \mathbb{R}^{D \times N}$ at the iteration $t$. $\mathbf{X}^t$ is the observation at time $t$ when an augmentation policy is applied to the existing data points $\mathbf{X}$.

Our model can be applied to a variety of DSC networks. In this section, we consider a general form that consists of an encoder that takes $\mathbf{X}^t$ as an input and generates latent space features $\mathbf{Z}^t$. The latent space features are reconstructed by a self-expressive layer with parameters $\mathbf{C}^t$. That is, $\mathbf{Z}^t\mathbf{C}^t$ is fed to the decoder to develop $\hat{\mathbf{X}}^t$, which is a reconstruction of $\mathbf{X}^t$. Figure 1 shows an overview of this model. Note that such a model includes a fully-connected layer that connects all the samples in the mini-batch (the self-expressive layer). Thus, the number of data points and their orders cannot be changed during the training. We keep a placeholder with $N$ fields that correspond to the existing samples and feed $\mathbf{X}^t$ to this placeholder at every training step $t$. The permutation of samples in $\mathbf{X}^t$ remains the same.

As mentioned, we aim for an autoencoder that preserves the subspace membership of slightly transformed inputs. Let $\mathbf{C}^t$ be the coefficient matrix that is constructed at the $t$-th iteration of a subspace clustering algorithm. In addition, let $\hat{\mathbf{Q}}$ be an existing estimation of subspace membership matrix, whose rows are one-hot vectors denoting the subspace memberships assigned to different samples. The multiplication of $\hat{\mathbf{Q}}^T$ and $|\mathbf{C}^t|$ gives a matrix whose $(i, j)$th element shows the contribution of the samples assigned to the $i$-th subspace in reconstructing the $j$-th sample. For a perfect subspace-preserving coefficient matrix, $\hat{\mathbf{Q}}^T|\mathbf{C}^t|$ has only one non-zero element in each row.

For each sample $j$, the maximum value in the $j$-th row of $\hat{\mathbf{Q}}^T|\mathbf{C}^t|$ can point to a new estimate for its subspace membership. Therefore, a prediction of subspace membership matrix at the iteration $t$ can be calculated as follows

$$\mathbf{Q}^t = \text{Softmax}(\hat{\mathbf{Q}}^T|\mathbf{C}^t|), \tag{4}$$

where Softmax($\cdot$) corresponds to the softmax function on the rows of its input. We refer to $\mathbf{Q}^t$ as *temporal* subspace membership matrix.

The temporal subspace membership matrix $\mathbf{Q}^t$ estimates the subspace memberships for the current observation $\mathbf{X}^t$. Note that because of the randomly augmented inputs, the coefficient matrix $\mathbf{C}^t$ can undergo sudden changes in different time frames. While it is fine to have different coefficient matrices for slight transformations of data, we are interested in maintaining persistent subspace membership matrices $\mathbf{Q}^t$. Thus, we propose a subspace membership consistency loss.

We keep an exponential moving average (EMA) of $\mathbf{C}^t$s, the coefficient matrices, to provide a smooth temporal ensemble for the coefficient matrix. Thus, in addition to the *temporal* subspace membership matrix in (4), in each training iteration, we can calculate another membership matrix corresponding to the temporal ensemble of coefficient matrices in prior iterations. We refer to this membership matrix as $\mathbf{Q}^t_{\text{Ens.}}$.

Let $\mathbf{C}^{t-1}_{\text{EMA}}$ be the EMA of coefficient matrices until the iteration $t-1$, and $\mathbf{C}^t$ be the calculated update for the coefficient matrix at the iteration $t$. The EMA of the coefficient matrix at the iteration $t$ can be updated as follows

$$\mathbf{C}^t_{\text{EMA}} = \alpha\mathbf{C}^{t-1}_{\text{EMA}} + (1-\alpha)\mathbf{C}^t, \tag{5}$$

where $0 < \alpha < 1$ is the smoothing factor. Using $\mathbf{C}^t_{\text{EMA}}$ we can calculate $\mathbf{Q}^t_{\text{Ens.}}$ as

$$\mathbf{Q}^t_{\text{Ens.}} = \text{Softmax}(\hat{\mathbf{Q}}^T|\mathbf{C}^t_{\text{EMA}}|), \tag{6}$$

where $\hat{\mathbf{Q}}$ is the same prior membership matrix as in (4).

Note that $\mathbf{Q}^t_{\text{Ens.}}$ provides more consistent subspace membership predictions as compared to $\mathbf{Q}^t$. To encourage the autoencoders to favor functions that preserve the subspace memberships even for differently transformed observations $\mathbf{X}^t$, we propose the subspace membership consistency loss as follows:

$$\mathcal{L}_{\text{Cons.}}(\mathbf{Q}^t_{\text{Ens.}}, \mathbf{Q}^t) = \text{CE}(\mathbf{Q}^t_{\text{Ens.}}, \mathbf{Q}^t), \tag{7}$$

where CE($\cdot$) denotes the cross-entropy function. $\mathcal{L}_{\text{Cons.}}$ penalizes the temporal changes to the subspace memberships if they are inconsistent with the temporal ensemble of subspace memberships $\mathbf{Q}^t_{\text{Ens.}}$.

**Full Objective.** We train the networks iteratively with two steps of subspace clustering and subspace membership consistency in each iteration. In the subspace clustering step, the loss function of the subspace clustering algorithm of choice (3) is optimized, and in the subspace membership consistency step, (7) is optimized. That is at each iteration $t$, we train the netwrok with the following algorithm.

$$\begin{cases} \text{Step 1:} & \min_\Theta(\mathcal{L}_{\text{S.E.}}(\mathbf{C}^t, \mathbf{Z}^t) + \lambda_1\mathcal{L}_{\text{S.P.}}(\mathbf{C}^t) + \lambda_2\mathcal{L}_{\text{Rec.}}(\mathbf{X}^t, \hat{\mathbf{X}}^t)), \\ \text{Step 2:} & \min_\Theta(\mathcal{L}_{\text{Cons.}}(\mathbf{Q}^t_{\text{Ens.}}, \mathbf{Q}^t)), \end{cases} \tag{8}$$

where $\Theta$ is the union of trainable parameters in the networks.

## 4 Finding Efficient Augmentations

In the previous section, we denoted $\mathbf{X}^t$ as a stochastic transition of $\mathbf{X}$ which is the result of applying an augmentation policy. The choice of augmentation policy plays an important role in the performance

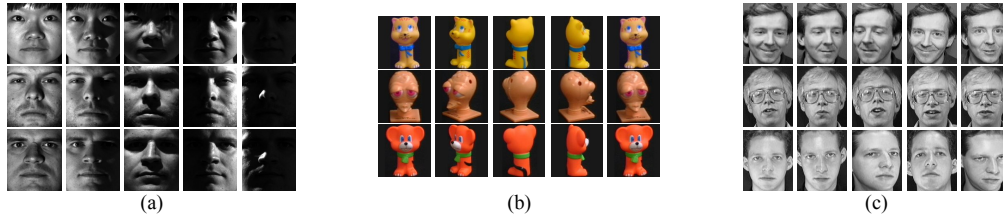

Figure 2: Sample images from different used datasets. (a) Extended Yale-B dataset [39]. (b) COIL dataset [40, 41] . (c) ORL dataset [42].

of the network. We formulate the problem of finding the best augmentation policy as a discrete search problem.

Our method consists of three components: A *score*, a *search algorithm* and a *search space* with $n_s$ possible configurations. The search algorithm samples a data augmentation policy $S_i$, which has information about what image processing operation to use, the probability of using the operation in each iteration, and the magnitude of the operation. The policy $S_i$ will be used to train a child deep subspace clustering network with a fixed architecture. The trained child network will return a score that specifies the effect of applying the policy $S_i$ to the input data on the performance of deep subspace clustering task. Finally, all the tested policies $\{S_i\}_1^{n_s}$ will be sorted based on the returned scores.

In the following, we describe the *score* , the *search algorithm* and the *search space* in detail.

**Score.** In our framework, the score is a metric that evaluates the performance of the DSC on a certain given input. Note that the ground-truth labels are unknown at this stage. Therefore, we need to use a validation technique that does not use the ground-truth labels. Any internal validation of clustering methods, including mean Silhouette coefficient [37] or the Davies-Bouldin index (DBI) [38] can serve as the score metric in our search. We use mean Silhouette coefficient in this paper.

**Search Space.** In our search space, a sample policy $S_i$ consists of $\ell$ sequential sub-policies with each sub-policy using an image operation. Additionally, each operation is also associated with two hyper-parameters: 1) the probability of applying the operation, and 2) the magnitude of the operation. We discretize the range of probability and magnitude values into $n_p$ and $n_m$ discrete values, respectively (with uniform spacing). This way, we can use a discrete search algorithm to find them. For $n_o$ operations, this constructs a search space with the size of $n_s = (n_o \times n_p \times n_m)^{\ell}$.

**Search Algorithm.** The size of search space $n_s$, can grow exponentially. A brute-force search might be impractical. To make the searching process feasible, we use a greedy search. First, we begin searching in the reduced search space where each sample policy has only one sub-policy ($\ell = 1$). In the reduced search space, we find the best probability and magnitude for each image operation. Note that $n_p$ and $n_m$ can also be decreased as much as necessary to keep the search tractable.

Once we find the best augmentation operations for the first sub-policy, we search for the second sub-policy ($\ell = 2$). For each found sub-policy in the first stage, we search for the best combination of image operations and their probabilities and magnitudes.

This process continues until we reach $\ell = \ell_{\max}$, the maximum number of sub-policies. At this point, we sort all the potentially good policies that are found until this point, and select the best $b$ augmentation policies among them.

## 5   Experimental Results

We evaluate our method against state-of-the-art subspace clustering algorithms on three standard datasets. We first use the algorithm described in section 4 to find the best augmentation policies for each dataset. Then, we use the found augmentation policies in the ablation study as well as in comparisons with state-of-the-art subspace clustering algorithms.

We use the following datasets in our experiments:

**Extended Yale-B dataset** [39] contains 2432 facial images of 38 individuals from 9 poses and under 64 illuminations settings.

**ORL** dataset [42] includes 400 facial images from 40 individuals. This corresponds to only 10 samples per subject.

**COIL-100** [40] and **COIL-20** [41] datasets are respectively consisted from images of 100 and 20 objects placed on a motorized turnable. Per each object, 72 images are taken at pose intervals of 5 degrees that covers a 360 degrees range. Following most of the prior studies, in our experiments, we use grayscale images of these datasets.

Figure 2 (a), (b), and (c) show sample images from the Extended Yale-B, ORL and COIL datasets, respectively. Note that in the subspace clustering tasks, the datasets are not split into training and testing sets. Instead, all the existing samples are used in both the learning stage and the performance evaluation stage.

**Experimental Setups.** While our method can be applied to many DSC algorithms, unless otherwise stated, due to its promising performance, we adopt the MLRDSC networks [17] and apply our method to its networks. We call the result MLRDSC with Data Augmentation (MLRDSC-DA). The objective function of MLRDSC can be also written in the format of (3). The *self-expressiveness* and *subspace-preserving* loss terms in MLRDSC are

$$\mathcal{L}_{\text{S.E.}}(\mathbf{C}, \mathbf{Z}) = \sum_{l=1}^{L} \|\mathbf{Z}_l - \mathbf{Z}_l(\mathbf{G} + \mathbf{D}_l)\|_F^2 \quad \text{and} \quad \mathcal{L}_{\text{S.P.}}(\mathbf{C}) = \|\mathbf{Q}^T|\mathbf{G}|\|_1 + \lambda_3 \sum_{l=1}^{L} \|\mathbf{D}_l\|_F^2, \quad (9)$$

where $L$ is the number of layers in the autoencoder, $\mathbf{Z}^l$ is the features at the $l$-th layer, and $\mathbf{C} = \mathbf{G} + \frac{1}{L}\sum_{l=1}^{L} \mathbf{D}_l$. The coefficient matrix in this model is calculated by the consistency matrix $\mathbf{G}$ and distinctive matrices $\{\mathbf{D}_l\}_{l=1}^{L}$. The distinctive matrices enforce subspace-preserving across different layers, and $\mathbf{G}$ captures the shared information between the layers.

In the training of MLRDSC-DA, we first pre-train the networks by performing the MLRDSC algorithm. Then, we continue training MLRDSC-DA for a few additional iterations until convergence with (8) as

$$\begin{cases} \text{Step 1:} & \min_\Theta \quad \sum_{l=1}^{L} \|\mathbf{Z}_l^t - \mathbf{Z}_l^t(\mathbf{G}^t + \mathbf{D}_l^t)\|_F^2 + \lambda_1 \|\mathbf{Q}^{t^T}|\mathbf{G}^t|\|_1 \\ & \qquad\qquad + \lambda_3 \sum_{l=1}^{L} \|\mathbf{D}_l^t\|_F^2 + \lambda_2 \|\mathbf{X}^t - \hat{\mathbf{X}}^t\|_F^2, \\ \text{Step 2:} & \min_\Theta \quad \text{CE}(\mathbf{Q}_{\text{Ens.}}^t, \mathbf{Q}^t), \end{cases} \quad (10)$$

where we shape the temporal coefficient matrix as $\mathbf{C}^t = \mathbf{G}^t + \frac{1}{L}\sum_{l=1}^{L} \mathbf{D}_l^t$, and $\mathbf{Q}_{\text{Ens.}}^t$ and $\mathbf{Q}^t$ are calculated from (6) and (4), respectively.

We use the same training settings as described in [17]. This includes the same architecture for networks and values for the hyper-parameters $\lambda_1$, $\lambda_2$, $\lambda_3$ in different experiments as well as the initial values of a zero matrix for the membership matrix $\hat{\mathbf{Q}}$, and matrices with all the elements equal to 0.0001 for the coefficient matrices $\mathbf{G}^0$ and $\mathbf{D}_l^0$s at the iteration $t = 0$. We update $\hat{\mathbf{Q}}$ every 50 iterations by substituting the subspace membership estimations with the result of subspace clustering performed on the current $\mathbf{C}^t$. We set the EMA decay to $\alpha = 0.999$ in all the experiments (selected by cross-validation and mean silhouette coefficient as the evaluation metric). We implemented our method with PyTorch. We use the adaptive momentum-based gradient descent method (ADAM) [43] with a learning rate of $10^{-3}$ to minimize the loss functions. Similar to other DSC methods, we input the whole dataset as a batch. In all the conducted experiments, we report 5-fold averages.

## 5.1 Best Augmentation Policies Found on the Datasets

We perform the search algorithm in Section 4 on different datasets to find the best augmentation policies for each dataset. To reduce the computations, we search in the search space of augmentation policies with the maximum number of sub-policies $\ell_{\max} = 2$ (i.e. up to two sub-policies can be combined to construct a policy), and set the probability to $p = 0.1$ and the magnitude to $m = 0.3 \times r$ where $r = (\max - \min)$ is the magnitude range that image operations accept. The image operation search space is the following set: {FlipLR, ShearX, FlipUD, SearY, Posterize, Rotate, Invert, Brightness, Equalize, Solarize, Contrast, TranslateY, TranslateX, AutoContrast, Sharpness, Cutout} that is also used in [21]. This results in a search space of $n_s = 16^2$. We selected the

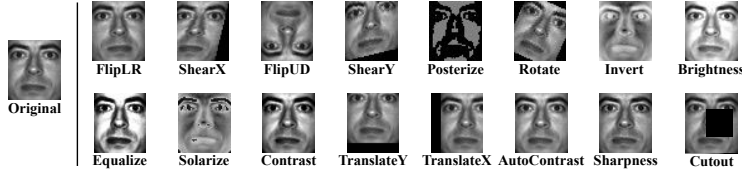

Figure 3: Different image transformations on a sample from the Extended Yale B dataset.

Table 1: Augmentation policies that yield the highest mean Silhouette coefficient in the subspace clustering results on different datasets.

| Dataset | Augmentation Policy 1 | Augmentation Policy 2 |
|---|---|---|
| Extended Yale B | (Op ='ShearY', m=0.3$r$, p=0.1) | (Op ='TranslateY', m=0.3$r$, p=0.1) + (Op ='Contrast', m=0.3$r$, p=0.1) |
| COIL-20 & COIL-100 | (Op ='Posterize', m=0.3$r$, p=0.1) + (Op ='Sharpness', m=0.3$r$, p=0.1) | (Op ='FlipLR', p=0.1) + (Op ='Contrast', m=0.3$r$, p=0.1) |
| ORL | (Op ='ShearX', m=0.3$r$, p=0.1) + (Op ='Sharpness', m=0.3$r$, p=0.1) | (Op ='FlipLR', p=0.1) + (Op ='ShearX', m=0.3$r$, p=0.1) |

Table 2: Ablation study of our method in terms of clustering error (%) on Extended Yale B. Top performers are bolded.

| Backbone | Augmentations × Consistency Loss × | Augmentations ✓ Consistency Loss × | Augmentations × Consistency Loss ✓ | Augmentations ✓ Consistency Loss ✓ |
|---|---|---|---|---|
| DSC | 2.67 | 3.10 | 2.56 | **1.92** |
| MLRDSC | 1.36 | 2.84 | 0.95 | **0.82** |

values for magnitude and probability of augmentation polices by searching in the full search space of augmentation policies for the first two subjects in the Extended Yale B dataset.

Figure 3 shows the different augmentation policies applied to a sample drawn from the Extended Yale B dataset. The details of these image operations are described in Table 1 in the supplementary materials.

For each candidate augmentation policy, we train our MLRDSC-DA model, perform subspace clustering, and return the mean Silhouette coefficient [37] as the clustering performance. We use the mean Silhouette coefficients to sort the augmentation policies(including policies with $\ell < \ell_{max}$ sub-policies) and select the top two performing augmentation policies in each dataset. That is $b = 2$.

Table 1 shows the found augmentation policies that yield to the highest Silhouette coefficients in the subspace clustering results on different datasets. In our experiments, COIL-20 and COIL-100 resulted in similar policies. Unless otherwise stated, in all the experiments, we apply these augmentation policies to the inputs of our MLRDSC-DA algorithm.

## 5.2 Ablation Study and Analysis of The Model

To understand the effects of some of our model choices, we explore some ablations of our model on the Extended Yale B dataset. In particular, we test our model on two different deep subspace clustering methods, DSC [15] and MLRDSC [17], and in four settings where 1) the consistency loss exists or 2) is ablated; 3) the optimal augmentations policies are applied to the inputs or 4) the data is fed without any augmentations.

If we remove both augmentations and the consistency loss, our networks, based on their backbones, turn to either DSC or MLRDSC networks. In the versions that data augmentation is applicable, the augmentations in Table 1 are used. Further analysis on the evaluation of the found augmentation policies is provided in section 5.4.

We report the performances in Table 2. As can be seen, the top performer is our full model with augmentations and the consistency loss applied to the MLRDSC method. MLRDSC-based methods, in general, outperform DSC-based methods. Consistency loss slightly improves the performance even without data augmentation. This is the result of temporal ensembling.

Table 3: Clustering error (%) of different methods on Extended Yale B, ORL, COIL20, and COIL100 datasets. Top performers are bolded.

| dataset | LRR [5] | LRSC [44] | SSC [45] | AE+SSC [15] | KSSC [9] | SSC-OMP [46] | EDSC [47] | AE+EDSC [47] | DSC [15] | DASC [16] | $S^2$ConvSCN [14] | MLRDSC [17] | MLRDSC-DA Ours |
|---|---|---|---|---|---|---|---|---|---|---|---|---|---|
| E.Yale B | 34.87 | 29.89 | 27.51 | 25.33 | 27.75 | 24.71 | 11.64 | 12.66 | 2.67 | 1.44 | 1.52 | 1.36 | **0.82** |
| ORL | 33.50 | 32.50 | 29.50 | 26.75 | 34.25 | 37.05 | 27.25 | 26.25 | 14.00 | 11.75 | 10.50 | 11.25 | **10.25** |
| COIL20 | 30.21 | 31.25 | 14.83 | 22.08 | 24.65 | 29.86 | 14.86 | 14.79 | 5.42 | 3.61 | 2.14 | 2.08 | **1.79** |
| COIL100 | 53.18 | 50.67 | 44.90 | 43.93 | 47.18 | 67.29 | 38.13 | 38.88 | 30.96 | − | 26.67 | 23.28 | **20.67** |

Table 4: Clustering error (%) on Extended Yale B with different augmentation policies applied to the inputs of MLRDSC-DA.

| Augmentation Policies: | Random LR Flips | Cut-out | Common aug. policies | AutoAug for ImageNet | AutoAug for SVHN | Policies found from Algorithm 1 (ours) |
|---|---|---|---|---|---|---|
| Extended YaleB | 1.32 | 2.88 | 2.96 | 5.96 | 11.31 | **0.82** |

As can be seen in the second column of this table, applying the found augmentations to the input of DSC and MLRDSC networks without further modification (i.e., not adding the consistency loss) not only does not improve the results, but it slightly degrades the performance. These results clearly show both the importance of the consistency loss and the benefit of using data augmentations when it is combined with the consistency loss.

## 5.3 Comparison with State-of-The-Art Subspace Clustering Methods

In this section, we evaluate our method against the state of the art subspace clustering methods. We apply the found augmentation policies in Table 1 to the data on Extended Yale B, ORL, COIL-20 and COIL-100 datasets and feed them to our MLRDSC-DA method.

The rows in Table 4 report the clustering error rates of different subspace clustering algorithms. As the table reveals, deep subspace clustering methods, including DSC, ADSC, $S^2$ConvSCN, and ML-RDSC, in general, outperform the conventional subspace clustering approaches. This observation suggests that deep networks can better model the non-linear relationships between the samples. However, among them, our model outperforms all the benchmarks. Note that our model and MLRDSC share similar architectures and have the same number of parameters. The only difference is that our method takes advantage of training on the augmented set of data. This observation clearly shows the benefits of incorporating data augmentation in the task of deep subspace clustering.

## 5.4 Comparison with Common Augmentation Policies and Transferred Augmentation Policies

Existing automated learning algorithms for finding proper augmentations or even manual searches do not apply to the subspace clustering task. The current algorithms are mostly designed for supervised tasks and require the ground-truth targets to compare the performances, whereas, in the subspace clustering task, the ground-truth labels are not available. However, one may apply the supervised augmentation searches to a source dataset with available labels and use the found augmentation policies on a target dataset for the task of subspace clustering.

To compare such an approach with the described method in Section 4, we adopt the augmentation policies that AutoAug [21] finds on the classification task for SVHN [48] and ImageNet [49] datasets, and directly apply the found policies to the input of our MLRDSC-DA.

We furthermore compare the performances to the results of applying the following augmentation policies to the input: random left-right flips (*Flip-LR*), *Cut-out* [50, 51] and common augmentations picked by practitioners (*Common aug. policies*). For "Common aug. policies", we use the combination of most common augmentations, including zero paddings, cropping, random-flips, and cutout.

Note that all the experiments in this section share the same architecture and training procedure as MLRDSC-DA. They are only different in the augmentation policies that are applied to their input.

As can be seen in Table 4, the augmentation policies that are found with [21] on SVHN and ImageNet, perform poorly. This is because they are deemed good policies for the classification task on those datasets and may not work as efficiently on the subspace clustering task. The reason that Random Flips provides a relatively good performance is that the objects in the dataset are symmetric. The augmentations that are found with our suggested approach provide the best results.

## 6 Conclusion

We introduced a framework to incorporate data augmentation techniques in Deep Subspace Clustering algorithms. The underlying assumption in subspace clustering tasks is that data points with the same label lie into the same subspace. Based on this assumption, we argued that slight transformations of a data point should not alter the subspace into which the data point lies. To address this property, we proposed the subspace consistency loss to keep the data points within consistent subspaces when slight random transformations are applied to the input data. Employing the mean Silhouette coefficient metric, we furthermore, provided a simple yet effective unsupervised algorithm to find the best augmentation policies for each target dataset. Our experiments showed that applying good data augmentations improves the performance oft subspace clustering algorithms.

## 7 Broader Impact

Since our method improves subspace clustering, it advances learning from unannotated data. Improving the learning process and providing more accurate similarity matrices for unannotated data can positively impact accountability, transparency and explainability of AI methods. However, if not controlled, providing the opportunity to learn from big unannotated datasets could increase the concerns about violating the privacy of individuals.

## 8 Acknowledgement

This research was funded in part by NSF grants IIS-1703883, CNS-1747778, CCF-1733843, IIS-1763523, IIS-1849238-825536 and MURI-Z8424104-440149, and in part by the Northrop Grumman Mission Systems Research in Applications for Learning Machines (REALM) initiative and the NSF grant 1618677.

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
