[Supplementary Material]

# Supplementary Materials : Deep Subspace Clustering with Data Augmentation

**Mahdi Abavisani**
Rutgers University
New Brunswick, NJ
mahdi.abavisani@rutgers.edu

**Alireza Naghizadeh**
Rutgers University
New Brunswick, NJ
ar.naghizadeh@rutgers.edu.

**Dimitris N. Metaxas**
Rutgers University
New Brunswick, NJ
dnm@cs.rutgers.edu

**Vishal M. Patel**
Johns Hopkins University
Baltimore, MD
vpatel36@jhu.edu

## Abstract

In this section, we provide the details of the image operations that were used in our search space for augmentation policies, a sensibility test for the EMA decay parameter $\alpha$ in our model, and our procedure of reducing the search space of augmentation policies is explained. Finally, we provide some additional experiments for evaluation of our proposed method.

## 1 Details of Image Operations Used in Our Augmentation Policies

As mentioned in Section 5.1, our image operation search space for augmentation policies is the following set: {FlipLR, ShearX, FlipUD, SearY, Posterize, Rotate, Invert, Brightness, Equalize, Solarize, Contrast, TranslateY, TranslateX, AutoContrast, Sharpness, Cutout}. Table 1 lists these transformation techniques. The description of the magnitude is shown in the second column. In addition, the ranges of the magnitudes are presented in the third column. There are some transformations such as Invert and Equalize that do not use the magnitude information.

## 2 EMA Decay Parameter

In all of our experiments, we used the training settings of MLRDSC described in [2] for our MLRDSC-DA model. This includes the same architecture for networks and values for the hyper-parameters $\lambda_1$, $\lambda_2$, $\lambda_3$ in different experiments as well as the initial values for the membership matrix $\hat{\mathbf{Q}}$, the coefficient matrices $\mathbf{G}^0$ and $\mathbf{D}_l^0$s at the iteration $t = 0$.

Our model introduces one additional regularization parameter to MLRDSC (the EMA decay $\alpha$ in (5)). In this section, we analyze the sensibility of the proposed method to determine the $\alpha$ value. Table 4 shows the effect of this parameter on the performance of the EMA decay on the Extended Yale-B dataset.

In Table 4, we test different value of $\alpha$ over the range of $[0.09, 0.9999]$. As can be seen from the table, higher values generally resulted in a better performance. This is inline with the choice of EMA decay value in previous works [3].

Table 1: List of the image transformations in our search space.

| Operation Name | Description | Magnitudes' range |
|---|---|---|
| ShearX(Y) | Shearing the image along the horizontal ( or vertical) axis with rate *magnitude*. | [-0.3,0.3] |
| TranslateX(Y) | Translating the image in the horizontal (or vertical) direction by *magnitude* number of pixels. | [-15,15] |
| FlipRL(UD) | Flipping in the right-left (Up-down) direction. | |
| Rotate | Rotating the image by *magnitude* degrees. | [-30,30] |
| AutoContrast | Maximizing the the image contrast, by making the darkest pixel black and lightest pixel white. | |
| Invert | Inverting the pixels of the image. | |
| Equalize | Equalizing the image histogram. | |
| Solarize | Inverting all pixels above a threshold value of *magnitude*. | [0,256] |
| Posterize | Reducing the number of bits for each pixel to *magnitude* bits. | [4,8] |
| Contrast | Controlling the contrast of the image. A *magnitude*=0 gives a gray image, whereas *magnitude*=1 gives the original image. | [0.1,1.9] |
| Brightness | Adjusting the brightness of the image. By the scaled *magnitude*, where *magnitude*=0 gives a black image, whereas *magnitude*=1 gives the original image. | [0.1,1.9] |
| Sharpness | Adjusting the sharpness of the image. A *magnitude*=0 gives a blurred image, whereas *magnitude*=1 gives the original image. | [0.1,1.9] |
| Cutout [1] | Setting a random square patch of side-length *magnitude* pixels to gray. | [0,6] |

Table 2: Clustering error (%) on Extended Yale B with different $\alpha$ values of EMA decay for MLRDSC-DA.

| The $\alpha$ Values: | $\alpha = 0.09$ | $\alpha = 0.9$ | $\alpha = 0.99$ | $\alpha = 0.999$ | $\alpha = 0.9999$ |
|---|---|---|---|---|---|
| Extended YaleB | 3.04 | 2.88 | 0.9 | **0.82** | 0.86 |

## 3 Reducing The Search Space of Augmentation Policies

As mentioned in Section 5.2, to reduce the computations, we used a greedy search [4] in the search space of augmentation policies with the maximum number of sub-policies $\ell_{\max} = 2$ (i.e. up to two sub-policies can be combined to construct a policy), and set the probability to $p = 0.1$ and the magnitude to $m = 0.3 \times r$ where $r = (\max - \min)$ is the magnitude range that image operations accept. The range of each image operation is shown in Table 1.

We selected the values $p = 0.1$ and $m = 0.3 \times r$, for magnitude and probability of augmentation polices by searching in a larger search space of augmentation policies for the first two subjects in the Extended Yale B dataset. In particular, we created a subset of Extended Yale B dataset with the first 128 samples corresponding to the first two subjects of the dataset. We ran the search algorithm for a range of different values for $p$ and $m$, and chose the median value of $p$ and $m$ that resulted in the best mean Silhouette scores across different image operations.

For both $p$ and $m$ we searched through values of [0.1, 0.2, ..., 1.0]. That means, for each probability, there are 10 number of magnitudes. In most of the different image operations, $p = 0.1$ and $m = 0.3$ resulted in better performances.

## 4 Additional Evaluations

**Runtime**: In Table 2, we report the runtime for the policy search algorithm on the COIL-20 dataset.

**Silhouette Coefficient v.s. Ground-truth**: We tested our augmentation algorithm with accuracy as the Score on the COIL-20 dataset, and observed that for a set of 9 best augmentation policies it achieves the same clustering error rate of 1.79. Our found augmentations in Table 1 of the main

paper are also among these best augmentation policies.

**Search Baselines**: In Table 3, we compare the clustering error rate of training our method with the augmentations found by 1) our algorithm, 2) uniform sampling, and 3) coordinate descent. We test on the Extended Yale-B dataset.

**Downstream tasks**: We use the latent space features from our network and from MLRDSC to perform a classification task. We randomly set $50\%$ of the learned latent space features for ORL samples as test set and use the remaining samples in training an SVM model. Table 4 shows that features learned via our method are better for classification.

**Training deeper networks**: We add two more layers to DSC network (denoted by DSC-5layer). Table 5 compares the clustering error rate of DSC-5layer with and without augmentations applied to the ORL dataset.

| Table 2: Runtime. | Table 3: Error rates on Yale-B. Search baselines. | | | Table 4: Downstream task: Accuracy of SVM on ORL. | | Table 5: Error rates on ORL. Deeper networks: DSC-5layer. | |
|---|---|---|---|---|---|---|---|
| COIL-20 | Ours | Uniform | Coordinate Descent | Ours | MLRDSC | Without Aug.s | With Aug.s |
| 261 mins | **0.82** | 3.72 | 0.95 | **95.5** | 93 | 22.50 | **13.25** |