[Reviews · NeurIPS 2020]

Review 1

Summary and Contributions: This paper introduces a framework for data augmentation in subspace clustering. The two main contributions of this work are (1) the design of a scheme to determine a data augmentation policy in an unsupervised way and (2) a temporal ensembling approach to maintain the consistency between subspace assignments across iterations. Experiments on 4 standard subspace clustering datasets demonstrate the effectiveness of the approach and its two components. I am happy with the rebuttal provided by the authors, who addressed the main points I raised, and I am positively updating the score.

Strengths: - Experiments are extensive and convincingly demonstrate the effectiveness of the approach compared to prior works. In addition to the comparison against state-of-the-art baselines, the ablation study and the comparison to common augmentation policies further support the claims of the paper and nicely complete the set of experiments. - The data augmentation scheme is flexible and thus any deep subspace clustering model could benefit from it. The experiments also confirm this by showing that improvements are obtained by applying the approach to both DSC and MLRDSC. - The paper reads very well and is clear for the most part.

Weaknesses: - The main weakness of the paper is the fact that efficiency is not discussed. Although several efforts were made to limit the size of the search space (reduced to 16^2=256 in the experiments), it is still unclear how costly it is to train the child network with each of these 256 policies. Covering the question of runtime (in particular for the policy search phase) is needed to assess the practicality of the approach. - I did not see any mention of the fact that the source code would be publicly released, should the paper be accepted. Yet, although such open-sourcing effort would be appreciated, this is not too detrimental to the work as the data augmentation scheme seems simple enough to be easily re-implemented.

Correctness: The claims are verified through extensive and convincing experiments. The methodology used in the experiments is also sound. However, the authors mention that EMA decay is to 0.999 through cross-validation. Given that no labeled data are available in the clustering task, more details on how this cross-validation was carried on would be appreciated. Additionally, I did not find any mention of the number of runs performed for each approach/dataset pair. Was only one run performed or do the results report the average over several runs? In the former case, how can one make sure that the results are not due to the randomness underlying the training process?

Clarity: The paper is well-written and clear for the most part. There are nonetheless some points I would like to see clarified: - Based on the explanation in Lines 113-117, I did not fully understand whether the model is trained with minibatches or with the whole dataset as batch. - Is there any intuition or rationale explaining why the consistency loss is not combined in the same loss as the subspace-expressiveness, subspace-preserving and reconstruction terms? In other words, why not combine Step 1 and Step 2 in Equation 8? - Are augmentation policies with strictly less than l_max sub-policies considered as candidates in the search algorithm? - In Equation 9, is Q^T the same as \hat{Q}^T? - Section 5.1 mentions that the two best policies found by the policy search algorithm are selected. However it is not fully clear how the two policies are used when the model is trained. Is a uniform sampling performed to choose between either of the two policies to augment the samples?

Relation to Prior Work: Previous deep subspace clustering works are properly covered and the difference with respect to data augmentation in the supervised settings was also discussed.

Reproducibility: Yes

Additional Feedback: - In the definition of the membership consistency loss, did the authors try to replace Q^t_Ens with \hat{Q}? It seems that doing so would be an alternative to enforce consistency in Q^t as \hat{Q} reflects the subspace membership assignments obtained a few iterations before. - Another set of experiments that would have been interested to conduct is to compare the policies selected with the mean Silhouette coefficient to the policies selected with a groundtruth-based clustering metric such as Clustering Accuracy (on a held-out validation set). This would show how the unsupervised setting influences the policy selection, and how much it would help to have some labels (in a semi-supervised case). - I would appreciate the addition of a 'broader impact' section to the paper, in particular to discuss the potential application of this work to facial recognition and its potential harms. - The paper contains some typos: * Line 23: This motives us => This motivates us * Lines 31, 32: which essentially which learn => which essentially learn * Line 40: as apposed => as opposed * Line 308: Additional analysis of are provided => Additional analyses are provided


Review 2

Summary and Contributions: The paper considers the problem of subspace clustering of images using deep networks. The main idea is to enforce consistency of subspace membership under data augmentations (e.g. flipping). The paper further proposes a greedy search method for finding the data augmentation policy. The method achieves consistent improvements in tested settings.

Strengths: 1. The idea is interesting and relevant to the community 2. The method achieves consistent improvements in tested settings 3. The method is tested across different datasets and subspace clustering algorithms 4. Different aspects of the method are ablated (e.g. the impact of augmentation and consistency loss)

Weaknesses: 1. The proposed policy method found via the proposed greedy search strategy results outperforms policies found in the fully-supervised setting of ImageNet classification (by AutoAugment and practitioners). However, it is hard to tell if a different search method would result a better policy. It would be good to include baselines for the search method. For example, a simple random search baseline (e.g. uniform sampling similar to [1]) and a coordinate descent baseline (classic manual search; e.g. as described in [2]). 2. The idea of enforcing consistency under different transformations of unlabeled data has been used in a number of works largely in the semi-supervised and self-supervised learning literature (e.g. [3, 4, 5, 6, 7, 8]). It would be good to discuss this. It would also be good to discuss related work on searching for data augmentation policies (e.g. [1, 9]) and general hyperparameter optimization. 3. The paper shows consistent improvements on subspace cluster task across several datasets. It would be nice to also show results on using the learnt features for a downstream tasks (e.g. similar to how features using via clustering are evaluated on downstream vision tasks [10]). 4. It would be good to test the method for different network architectures (e.g. bigger networks) and training schedules (e.g. longer training) to see how the method generalizes to such settings. References: [1] Cubuk et al, RandAugment: Practical automated data augmentation with a reduced search space, 2019 [2] Larochelle et al, An empirical evaluation of deep architectures on problems with many factors of variation, ICML 2017 [3] Bachman et al, Learning with pseudo-ensembles, NIPS 2014 [4] Sajjadi et al, Regularization with stochastic transformations and perturbations for deep semi-supervised learning, NIPS 2016 [5] Laine and Aila, Temporal Ensembling for Semi-Supervised Learning, ICLR 2016 [6] Simon et al, Hand keypoint detection in single images using multiview bootstrapping, CVPR 2017 [7] Radosavovic et al, Data Distillation: Towards Omni-supervised Learning, CVPR 2018 [8] Tian et al, Contrastive Multiview Coding, 2019 [9] Ho et al, Population Based Augmentation: Efficient Learning of Augmentation Policy Schedules, ICML 2019 [10] Caron et al, Deep Clustering for Unsupervised Learning of Visual Features, ECCV 2018

Correctness: The method is generally well-evaluated. However, it would be good to address the relevant points from the weaknesses section, (1) in particular.

Clarity: The paper is well-written.

Relation to Prior Work: The paper discusses related work on subspace clustering. However, the discussion of the related work on enforcing consistency for unlabeled data is missing. Point (2) from the weakness sections.

Reproducibility: Yes

Additional Feedback: Updated review: I thank the authors for the response. The rebuttal addresses my concerns regarding missing search baselines and evaluation on downstream tasks. Thus, I update my score.


Review 3

Summary and Contributions: This paper proposes to incorporate augmentation in deep subspace clustering(DSC). The contributions include: 1) A consistency loss for the subspace assignment within augmented versions of a data point. 2) A discrete search method that finds an efficient set of augmentations in terms of operations and their magnitudes.

Strengths: Data augmentations have been shown to be an effective approach in unsupervised learning. Makes sense to integrate it in DSC.

Weaknesses: Unfortunately, I could not fully follow the motivations behind the proposed contributions. The authors point out in the introduction that data augmentation is an effective technique in supervised learning to prevent from overfitting. This motivation does not hold in their context, as their dataset is unlabeled, and could be extended with remarkably lower cost. One can argue that incorporating augmentation helps with achieving higher quality subspace clustering, which is not discussed by the authors. I do not see a rational behind the augmentation search approach either. Is the goal to minimize the main objective function, i.e. Eq. 8? If so, a clear shortcut for this would be not to apply any augmentations(achieving zero loss with the second term in step 2). The paper does not include the Border Impact section.

Correctness: As I mentioned above, I am not fully convinced with the motivation of this work.

Clarity: I did not have any difficulties with following the paper. As a minor issue, the introduction suffers from discontinuity.

Relation to Prior Work: It is acceptable.

Reproducibility: Yes

Additional Feedback: The authors would be great to clarify the motivations behind the proposed contributions. ------------------------------------------ Post rebuttal: ------------------------------------------ The authors rebuttal addressed some of my concerns.


Review 4

Summary and Contributions: This paper proposed a method to find optimal data augmentation for deep subspace clustering. The optimal augmentation is discovered by searching a pre-defined sequence of operations with maximal score, e.g. maximal Silhouette coefficient. A temporal momentum mechanism is further introduced to regularize the clustering estimation with different augmentations.

Strengths: The data augmentation issue is introduced for deep subspace clustering. The proposed method achieved SOTA performance on several traditional subspace clustering datasets.

Weaknesses: How does this method differ from deep metric learning. Since the random augmentation has already violated the linear subspace assumption, it no longer makes sense to still enforce a linear subspace even in the latent space. Furthermore, there is no comparison with or discussion on the SOTA unsupervised clustering/constrastive learning approaches, e.g. [1,2]. Both methods also adopted momentum encoders. The augmentation search algorithm is very expensive as it involves retraining the whole model with each sampled augmentation operations. The found operations do not generalize new datasets either. The augmentation search optimizes a score, e.g. the Silhouette coefficient. Since the score is totally unsupervised, I suspect whether a higher score would necessarily mean better performance in clustering task. The description bewteen p3 line120:126 may not be correct. If C^t is an N*N matrix and Q is an N*C matrix, Q^tC is a C*N matrix. Then I don't see Q^tC should have one none-zero element in each row, but instead should be each column. p1 l31: which essentially which --> which essentially p2 l40: apposed --> opposed [1] Li, Junnan, et al. "Prototypical Contrastive Learning of Unsupervised Representations." arXiv preprint arXiv:2005.04966 (2020). [2] He, Kaiming, et al. "Momentum contrast for unsupervised visual representation learning." Proceedings of the IEEE/CVF Conference on Computer Vision and Pattern Recognition. 2020.

Correctness: The method is correct to the best of my knowledge.

Clarity: The paper is ok in writing.

Relation to Prior Work: Some discussion on more recent unsupervised deep clustering and contrastive learning is not covered in this paper.

Reproducibility: Yes

Additional Feedback: The author should provide more discussion on the relation to [1] and [2]. The author should also explain why higher silhouette coefficient necessarily means higher clustering performance. Because it is a purely measure of clustering structure rather the correctness of clustering assignment. After reading the rebuttal, I am still not fully convinced by the linear subspace assumption after augmentation. Therefore I will keep my initial rating.

[Author Response · NeurIPS 2020]

We sincerely thank the reviewers for their valuable comments. We proofread and fixed the mentioned errors.

**Broader Impact**: Since our method improves subspace clustering, it advances learning from unannotated data.
Improving the learning process and providing more accurate similarity matrices for unannotated data can positively
impact accountability, transparency and explainability of AI methods. However, if not controlled, providing the
opportunity to learn from big unannotated datasets could increase the concerns about violating the privacy of individuals.

**Related Work**: Thank you for the additional references. We will include and discuss them in the revised version.

**Publishing codes**: Upon the acceptance of our paper, we will publicly release the source codes.

**Equation 9**: While in MLRDSC (Eq. (9)), $Q^T$ is the same as $\hat{Q}^T$, in our method, $Q^T$ relates to $\hat{Q}^T$ by Eq. (4).

**Replacing $Q_{Ens}^t$ with $\hat{Q}$** in Eq. (7) could also make an alternative consistency loss. However, using $Q_{Ens}^t$ has the
advantages of: 1) Ensembling multiple subspace membership predictions which results in a better estimate. 2) Providing
smoother transitions for subspace membership predictions that could result in better learning.

**Motivations behind augmentations for subspace clustering**: Additional unlabeled data can only improve the repre-
sentation learning in encoder and decoder, and not necessarily the self-expressiveness layer's parameters (which scales
quadratically with additional data). Augmentations in our model regularize parameters in both encoder and decoder
as well as the self-expressiveness layer. Intuitively, if two samples are similar, our method makes sure they remain
similar even if they undergo transformations that do not change their label. Adding additional data does not provide
such property.

**Rationale Behind the Augmentation Search**: Note that our augmentations search algorithm does not minimize
Eq. (8). It looks for augmentation policies that yield the highest mean Silhouette coefficient. The rationale is similar
to supervised models. Augmentations may initially increase the value of the loss function, but they lead to improved
learning by regularizing the training.

**Relation to metric learning and stochastic subspace clustering**: Our model shares more similarities to stochastic
subspace clustering (reference [17] in the submission) models (SSSC) than to metric learning methods. In SSSC,
stochastic transformations are applied to the inputs of the subspace clustering algorithm. In addition, many subspace
algorithms apply nonlinear functions to deal with nonlinearity (references [8,9,10,11] in the submission). We add the
discussion in the final paper.

**Silhouette coefficient**: Silhouette coefficient measures how similar an object is to its own cluster compared to other
clusters. As we assume in a clustering task we do not have any labels, an *external evaluation* such as Silhouette can be
used as an estimation of clustering performance to avoid bad augmentations.

**Experiments**: We include the following experiments in the revision:

• **Runtime**: In Table 1, we report the runtime for the policy search algorithm on the COIL-20 dataset.
• **Silhouette Coefficient v.s. Ground-truth**: We tested our augmentation algorithm with accuracy as the Score on the
COIL-20 dataset, and observed that for a set of 9 best augmentation policies it achieves the same clustering error rate
of 1.79. Our found augmentations in Table 1 of the main paper are also among these best augmentation policies.
• **Search Baselines**: In Table 2, we compare the clustering error rate of training our method with the augmentations
found by 1) our algorithm, 2) uniform sampling, and 3) coordinate descent. We test on the Extended Yale-B dataset.
• **Downstream tasks**: We use the latent space features from our network and from MLRDSC to perform a classification
task. We randomly set $50\%$ of the learned latent space features for ORL samples as test set and use the remaining
samples in training an SVM model. Table 3 shows that features learned via our method are better for classification.
• **Training deeper networks**: We add two more layers to DSC network (denoted by DSC-5layer). Table 4 compares
the clustering error rate of DSC-5layer with and without augmentations applied to the ORL dataset.

| Table 1: Runtime. | Table 2: Error rates on Yale-B. Search baselines. | | | Table 3: Downstream task: Accuracy of SVM on ORL. | | Table 4: Error rates on ORL. Deeper networks: DSC-5layer. | |
|---|---|---|---|---|---|---|---|
| COIL-20 | Ours | Uniform | Coordinate Descent | Ours | MLRDSC | Without Aug.s | With Aug.s |
| 261 mins | **0.82** | 3.72 | 0.95 | **95.5** | 93 | 22.50 | **13.25** |

**Details**: We include the following details in the revision:

• **EMA decay**: We used the silhouette coefficient as an evaluation metric in cross-validation.
• **Statistical Significance**: In all the conducted experiments, we reported 5-fold averages.
• **Minibatchs**: Similar to other DSC methods, we input the whole dataset as a batch.
• **Policies with $< \ell_{max}$ sub-policies** were also considered as augmentation candidates.

[Meta-Review · NeurIPS 2020]

This paper introduces a framework for data augmentation in subspace clustering. The method is well explained and back by thorough and convincing experiments. The main drawback of this work is that it lacks motivations to justify the need for this method, and this may greatly diminish the impact of the work if not clarified in the next iteration.